# Few-Shot Preference Learning for Human-in-the-Loop RL

**Joey Hejna**
Stanford University
jhejna@cs.stanford.edu

**Dorsa Sadigh**
Stanford University
dorsa@cs.stanford.edu

**Abstract:** While reinforcement learning (RL) has become a more popular approach for robotics, designing sufficiently informative reward functions for complex tasks has proven to be extremely difficult due their inability to capture human intent and policy exploitation. Preference based RL algorithms seek to overcome these challenges by directly learning reward functions from human feedback. Unfortunately, prior work either requires an unreasonable number of queries implausible for any human to answer or overly restricts the class of reward functions to guarantee the elicitation of the most informative queries, resulting in models that are insufficiently expressive for realistic robotics tasks. Contrary to most works that focus on query selection to *minimize* the amount of data required for learning reward functions, we take an opposite approach: *expanding* the pool of available data by viewing human-in-the-loop RL through the more flexible lens of multi-task learning. Motivated by the success of meta-learning, we pre-train preference models on prior task data and quickly adapt them for new tasks using only a handful of queries. Empirically, we reduce the amount of online feedback needed to train manipulation policies in Meta-World by $20\times$, and demonstrate the effectiveness of our method on a real Franka Panda Robot. Moreover, this reduction in query-complexity allows us to train robot policies from actual human users. Videos of our results can be found at https://sites.google.com/view/few-shot-preference-rl/home.

**Keywords:** Preference Learning, Interactive Learning, Multi-task Learning

## 1 Introduction

The success of deep reinforcement learning (RL) methods in game-playing and simulated domains [1] has inspired recent work applying RL-based techniques to real-world robot control to middling success. Integral to the success of deep RL methods is the reward function, which describes the desired behavior of the learning agent. While training robots via trial and error holds great promise, designing suitable reward functions remains challenging. For example, consider teaching a robot to open a door. The simplest reward function would be sparse – providing the robot with a positive reward only when the door has been opened. However, such sparse signals offer little learning signal, hampering exploration and enlarging sampling complexity. Conversely in designing a dense reward function, practitioners are tasked with summarizing multiple objectives like door angle or proximity to the handle into a single scalar. Such reward functions have proven to be difficult to design [2] and can even cause agents to learn unintended behaviors. Hand-designed dense reward functions often do not directly parallel the goal-conditions humans want them to capture, causing RL agents to exploit them and potentially leading to hazardous policies that do not align with human intent [3]. All of these problems are exacerbated in more realistic, multi-task scenarios with large state and action spaces [4] where we might wish to teach agents how to complete a variety of tasks in their environment. A robot that can only open doors provides little utility in the real world. Given the effort required to design a single reward function, constructing reward functions for an entire family of tasks is impractical.

Recent works attempt to circumvent the basic challenges of reward design by learning reward functions directly from human preferences. This paradigm has numerous advantages: learned

6th Conference on Robot Learning (CoRL 2022), Auckland, New Zealand.

reward functions are dense [5, 6], easily aligned with human intent [7], and can be adapted [8]. While demonstrations are often difficult to provide due to expensive data collection [9] and large domain gaps [10, 11], human preferences can often be elicited solely through simple pairwise comparisons. However, given the large continuous state and action settings of robotics problems, learning a high-performance reward function from only a handful of noisy user generated binary labels seems hopeless [12]. Consequently, methods from active learning maximize feedback efficiency by attempting to ask the most informative queries with simplistic or linear reward models [13, 14]. The constraints these methods place on the reward function class make them unable to scale to complex domains that necessitate expressive reward models [15]. Moreover, such methods are not significantly more data efficient than random sampling in practice [16, 17]. On the other hand, recent works using general function approximators still require thousands to tens-of-thousands of artificially labeled queries to learn sufficiently accurate reward functions [18, 19, 15]. This is far too onerous for real human labelers to provide, even in the single task setting. In order to train effective reward functions from actual humans, we need need a paradigm shift. Instead of optimizing for the most informative query, we take an orthogonal perspective that maximizes the amount of overall data by leveraging pre-training on realistic multi-task settings, and fine-tuning on a small and manageable amount of human queries online.

In the multi-task setting, significantly more data is available from previously known tasks which can be used to accelerate reward function learning. In fact, the shared structure of many real-world tasks has already been shown to accelerate policy learning [20]. The same structure can be exploited to learn complex reward functions for new tasks with only a handful of queries. This is largely because most tasks have rewards that are non-trivial compositions of other tasks. For example, data collected on opening windows and drawers could help us learn a reward function for door-opening with fewer human queries. Our key insight is to use multi-task data in order to meta-learn reward functions for preference based RL. Pre-training reward functions on a large dataset enables them to quickly adapt to new preferences with only a handful of queries.

Our core contributions are as follows. First we introduce a method for efficiently training RL policies from human-feedback using a meta-learned reward function. Second, we demonstrate its effectiveness across a number of standard robotics benchmarks, reducing query usage by a factor of 20 on robotic benchmarks in comparison to previous state-of-the-art methods. This increase in efficiency allows us to learn manipulation policies from real human feedback unlike prior work. Finally, we demonstrate the effectiveness of our method in the real-world using a Franka Panda robot.

## 2 Related Work

Our work builds on top of a number of prior works spanning RL, preference-learning, and meta-learning. Here we review the areas most relevant to our method.

**Reward Learning.** As hand-designed reward functions are difficult to tune, easily mis-specified [3, 21], and challenging to implement in the real world [2, 22], many recent works have leveraged human-collected data in order to learn reward functions. A large body of work focuses on using inverse RL, where a reward function is learned from approximately expert human collected demonstrations [23, 24, 25, 26]. However, demonstration collection is often expensive [27, 9, 28, 10, 29] and collected demonstrations are sometimes not even aligned with true human preferences [30, 31, 32]. Alternative strategies for learning reward functions utilize physical corrections [33], natural language instructions [34], human-provided scalar scores [35, 36] or partial [37] or complete [38, 39] rankings . While physical corrections and language may be easier for the user, it is generally unclear how they translate to reward updates. Stronger signals are provided by scalar scores or multiple rankings, but they are harder for users to provide [40]. We thus use pairwise comparisons as they are the simplest and generally refer to this approach as *preference learning*. Many recent works have studied active preference-based learning from human feedback, however such approaches often make restrictive assumptions of the reward function, like linearity in predefined features [13, 41, 42, 14, 43]. These assumptions make such methods too inexpressive to scale to modern robot learning with complex objectives [14]. While recent methods combining preference learning with deep RL make no assumptions on the structure of the reward function, they are far too feedback inefficient to be effectively used by humans [15, 18, 44, 45, 46]. Other works that use preferences with deep imitation learning [47] still require demonstrations. Most related to our work, PEBBLE [18] combines the SAC off-policy RL algorithm [48] with an ensemble

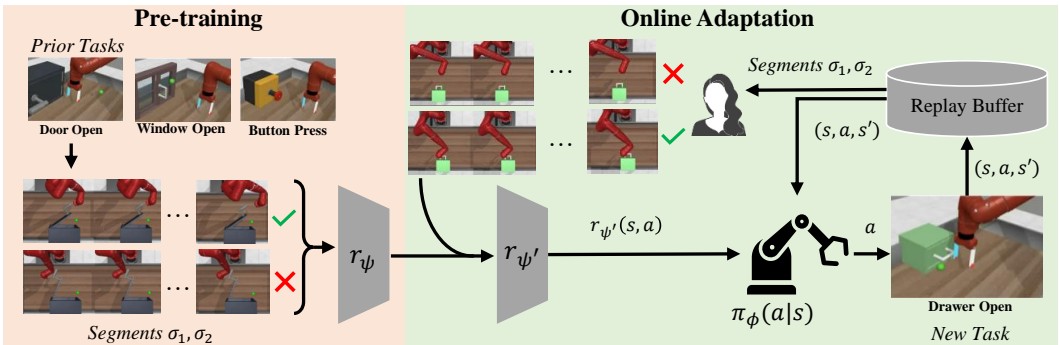

Figure 1: An overview of our method. **Pre-training (left):** In the pre-training phase we generate trajectory segment comparisons using data from a family of previously learned tasks and use them to train a reward model. **Online-Adaptation (Right):** After pre-training the reward model, we adapt it to new data from human feedback use it to train a policy for a new task in a closed loop manner.

of learned reward functions for sampling informative comparisons. Unfortunately, PEBBLE still requires an impractical number of queries to learn just a single task (25k for drawer opening). Distinct from prior work, we consider the more realistic multi-task setting that enables us to tap into a large amount of diverse data for for pre-training to increase query-efficiency.

**Meta Learning**. Meta-learning methods [49, 50] address the few-shot learning problem, where predictions on new tasks are made with a limited amount of data. Inspired by their success in supervised learning problems, we adopt the MAML algorithm [49] for learning new reward functions based on a limited number of human queries. Though supervised meta-learning has been previously used to infer reward classifiers [8] or in learning from heterogeneous demonstrators [51] to our knowledge it has not been applied to reward learning from preferences. Instead of adapting the reward function to new tasks, other related work in meta-RL directly adapts the policy network after a few exploratory episodes [52, 53, 54, 55]. As the RL problem is much more difficult than supervised reward learning, policy adaptation approaches are likely to be less query efficient.

## 3 Few-Shot Preference Learning for RL

In this section we formally describe the problem of meta-learning for preference based RL, then detail how our algorithm leverages multi-task pre-training for online few-shot adaptation.

**Problem Setup.** In standard RL, an agent maximizes its cumulative expected reward in a Markov decision process (MDP). Unlike standard RL, we assume the reward function $r(s, a)$ to be unknown and instead must be estimated from human feedback. Distinct from prior works in preference based RL, we focus on the multi-task regime and thus additionally assume the existence of a distribution of tasks $p(\mathcal{T})$. Each task $\tau$ corresponds to a unique MDP where the state space $\mathcal{S}$, action space $\mathcal{A}$, and discount factor $\gamma$ are held constant, but the unknown ground-truth reward function $r(s, a)$ and sometimes transition function $\mathcal{P}$, vary. Thus, we write that $\tau_i = (\mathcal{P}_i, r_i) \sim p(\mathcal{T})$.

Within this setting, we define the *few-shot preference-based RL* problem. Given access to a dataset of $N$ previous tasks, $\{\tau_i\}_{i=1}^N$, the agents goal is to learn a policy $\pi_{\text{new}}(a|s)$ for a new task $\tau_{\text{new}} \sim p(\mathcal{T})$ from human feedback with as few user queries as possible. We make no explicit assumption on the form of prior data for each of the $N$ prior tasks, only that it contains sufficient information to learn an estimate of the reward $r_i$. After the pre-training phase, depicted in the left half of Figure 1, we learn policies from online human feedback (right half of Figure 1). This setting is a significant departure from past work in preference-based RL, as we do not assume that new tasks are learned in isolation. More realistically, there are multiple tasks that have been completed within the same state and action space. Next we explain the major components of our approach.

**Preference Learning.** In order to learn the policy $\pi_{\text{new}}(a|s)$ for a new task from human preferences, we choose to learn the new tasks' reward function $r_{\text{new}}(s, a)$. While alternative approaches might seek to directly adapt the policy $\pi \rightarrow \pi_{\text{new}}$ using human feedback, such meta-RL style approaches often entail the difficult optimization challenges known to plague policy gradients and dynamic programming [56]. Instead, we directly model the reward using supervised learning techniques. We

denote $\hat{r}_\psi(s,a)$ to be a learned estimate of an unknown ground-truth reward function $r(s,a)$, parameterized by $\psi$. As in Wilson et al. [57] we consider preferences over partial trajectory segments $\sigma = (s_t, a_t, s_{t+1}, a_{t+1}, ..., s_{t+k-1}, s_{t+k-1})$ of $k$ states and actions, as they provide more information than single states [57, 15]. We then define a preference predictor over segments using the Bradley-Terry model of paired comparisons [58]:

$$P[\sigma_1 \succ \sigma_2] = \frac{\exp \sum_t \hat{r}_\psi(s_t^1, a_t^1)}{\exp \sum_t \hat{r}_\psi(s_t^1, a_t^1) + \exp \sum_t \hat{r}_\psi(s_t^2, a_t^2)}$$

In the above, $\sigma_1 \succ \sigma_2$ indicates the event that segment 1 is preferred to segment 2, as shown in Figure 1. For a given dataset $\mathcal{D}$ comprised of labeled queries $(\sigma_1, \sigma_2, y)$ where $y = \{1, 2\}$ corresponds to whether $\sigma_1$ or $\sigma_2$ is preferred, we optimize the following objective to learn $\hat{r}_\psi$.

$$\mathcal{L}_{\text{pref}}(\psi, \mathcal{D}) = -\mathbb{E}_{(\sigma^1, \sigma^2, y) \sim \mathcal{D}} \left[ y(1) \log(P[\sigma_1 \succ \sigma_2]) + y(2) \log(1 - P[\sigma_1 \succ \sigma_2]) \right] \quad (1)$$

In practice, this is just the standard binary cross-entropy objective where logits are determined by the sum of the learned reward function $\sigma$ over $k$ timesteps. Intuitively, this objective seeks to maximize the logits, and consequently predicted reward values, of the preferred segment in comparison to the unpreferred one.

**Pre-training for Preference Learning.** To estimate the reward function of a new task $r_{\text{new}}$ in as few queries as possible, we want to pre-train a reward function $\hat{r}_\psi$ that can quickly adapt to new tasks with only a handful of comparisons $(\sigma_1, \sigma_2, y)$. Tapping into offline data can help exploit shared task structure and potential accelerate learning on new tasks. We propose extending the meta-learning framework to preference learning across different tasks. Our approach is agnostic to the choice of meta-learning algorithm, but we choose Model Agnostic Meta-Learning (MAML) [49] for its simplicity. Concretely, MAML searches for parameters $\psi$ that attain high performance on a new task after only a few gradient steps by training on a set of previous tasks. In our setting, data for previous tasks can come from offline datasets, simulated policies, or actual humans. In conjunction with our preference loss from Equation (1), we use the following pre-training update:

---

**Algorithm 1** Few-Shot Preference-based RL

**Require:** Teacher freq $K$, Queries per session $M$
1: $\psi \leftarrow \arg\min_\psi \sum_i \mathcal{L}(\psi - \alpha \nabla_\psi \mathcal{L}(\psi, \mathcal{D}_i), \mathcal{D}_i)$
2: **for** $t = 1, 2, 3, ...$ **do**
3:     **if** $t\%K == 0$ **then**
4:         **for** $m = 1, 2, ...M$ **do**
5:             $(\sigma_1, \sigma_2) \sim$ Disagreement
6:             $y \leftarrow$ user preference
7:             $\mathcal{D}_{\text{new}} \leftarrow \mathcal{D}_{\text{new}} \cup (\sigma_1, \sigma_2, y)$
8:         **end for**
9:         $\psi' \leftarrow \psi$ Re-initialize reward model
10:         **for** each gradient step **do**
11:             $\psi' \leftarrow \psi' - \alpha \nabla_{\psi'} \mathcal{L}_{\text{pref}}(\psi', \mathcal{D}_{\text{new}})$
12:         **end for**
13:     **end if**
14:     Collect $s_{t+1}$ by taking $a_t \sim \pi(a_t|s_t)$
15:     Store transition $\mathcal{B} \leftarrow \mathcal{B} \cup (s_t, a_t, s_{t+1})$
16:     Sample batch $\{(s_t, a_t, s_{t+1})\}_{j=1}^B \sim \mathcal{B}$
17:     Assign rewards $r_t \leftarrow r_{\psi'}(s_t, a_t)$
18:     Optimize $\pi$ via SAC with
        $\{(s_t, a_t, s_{t+1}, r_{\psi'}(s_t, a_t))\}_{j=1}^B$
19: **end for**

---

$$\psi \leftarrow \psi - \beta \nabla_\psi \sum_{i=1}^N \mathcal{L}_{\text{pref}}(\psi - \alpha \nabla_\psi \mathcal{L}_{\text{pref}}(\psi, \mathcal{D}_i), \mathcal{D}_i). \quad (2)$$

Here $\alpha$ and $\beta$ are the inner and outer learning rates respectively. Each dataset $\mathcal{D}_i$ is comprised of known queries for each of the $N$ tasks $\tau_i \sim p(\mathcal{T})$. When we start training for a new task, we can quickly adapt the reward function using the new queries as $\psi' \leftarrow \psi - \alpha \nabla_\psi \mathcal{L}_{\text{pref}}(\psi, \mathcal{D}_{\text{new}})$. As $\psi$ is explicitly optimized for performance on $\mathcal{L}_{\text{pref}}$ after only a handful of updates, we significantly reduce query complexity.

Training $\hat{r}_\psi$ using Equation (2) however, requires access to query datasets $\mathcal{D}_i$ for each task. While pre-training can be accomplished through several objectives, like reward regression, we use preference-based pre-training for consistency and its generality. Pairwise comparison data can be extracted from a wide variety of sources. If reward values are present in offline data, artificial labels $y$ for trajectory segments $\sigma_1, \sigma_2$ can easily be generated via the comparison $\sum_t r(s_t^1, a_t^1) > \sum_t r(s_t^2, a_t^2)$ as is common practice in prior works [15, 18]. If reward values for previous tasks are unknown but policies are, reward values can be recovered via inverse-RL, or comparisons can be derived from direct behavior comparison. For example, when generating queries for task $i$, behaviors from $\pi_i(a|s)$ would be preferred to behaviors generated from $\pi_{\neq i}(a|s)$. The left half of Figure

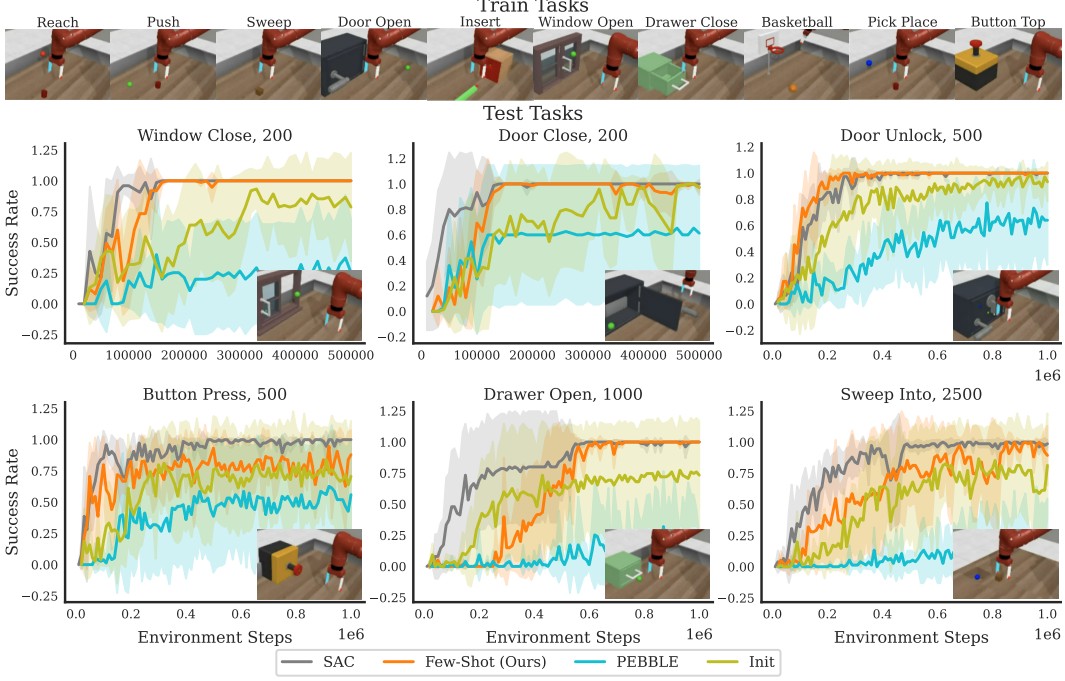

Figure 2: Results on MetaWorld tasks. The title of each subplot indicates the task and number of artificial feedback queries used in training. Results for each method are shown across five seeds.

1 shows the process of extracting query data from offline data for pre-training, which corresponds to line 1 in Algorithm 1. In our experiments, we use the artificial reward labeling scheme described first for consistency with prior work [15].

**Few-Shot Preference-based RL**. Our pre-trained preference function can then be used for few-shot preference based RL during an online adaptation phase, depicted in the right half of Figure 1. We modify the standard Soft-Actor Critic RL algorithm [48] to relabel transitions using our learned reward function before performing a standard actor-critic update (Algorithm 1 lines 17-18). Every $K$ steps, we ask a user to answer queries and provide feedback labels $y$ as shown in lines 5-7 of Algorithm 1. Informative queries are selected using the disagreement of an ensemble of reward functions over the preference predictors. Specifically, comparisons that maximize $std(P[\sigma_1 \succ \sigma_2])$ are selected each time feedback is collected [59]. After new feedback is collected, we re-initialize the reward model $\hat{r}_\psi$ to its pre-trained weights. Subsequently, we re-adapt it using the updated dataset $\mathcal{D}_{new}$ for the new task as shown in Algorithm lines 9-11.

To our knowledge, we are the first to leverage multi-task data for preference-based RL. The shift to the multi-task setting necessitates critical algorithmic changes in comparison with prior work. First, we pre-train the reward function from prior data instead of using other warm-start methods like unsupervised exploration used in PEBBLE. Second, we crucially reset the reward model for adaptation. Our setting provides a novel framework that leverages pre-training on a range of tasks for data-efficient adaptation on new tasks enabling human users to provide this data without making any structural assumptions on the reward function.

## 4    Experiments

In this section we seek to answer the following questions: First, does few-shot preference learning improve the query efficiency of preference-based RL? Second, is our method efficient enough to learn robot policies from real human feedback? Finally, can few-shot preference learning be used in the real world? Dataset, architecture, and hyperparameter details are available in the Appendix.

### 4.1 How query-efficient is few-shot preference-based RL?

To test the query-efficiency of few-shot preference based RL for realistic robotic tasks, we adopt the Meta-World benchmark from Yu et al. [20]. Agent tasks include household activities like opening doors or closing windows, and standard manipulation problems like block pushing. Some Meta-World tasks are particularly difficult for human-in-the-loop learning as they are sequential: feedback on the second part of the task, like where an agent should move a block, can only be provided once the agent learns the first part of the task, like how to grasp a block. Additionally, different objects introducing different manipulation dynamics across tasks. To evaluate the raw-performance of our approach, we use the artificial queries induced by the task ground truth reward function. Previous works in preference based RL have required up to fifty-thousand artificial queries in order to solve some of the Meta-world tasks [18]. Our approach generally achieves the same performance using $20\times$ fewer queries. Our reward models are pre-trained using *only 10 prior tasks* and evaluate query-efficiency on six previously unseen tasks. We compare our method, which we refer to as *Few-Shot*, to three baselines:

1. **SAC**: The Soft-Actor Critic RL algorithm trained from ground truth rewards. This represents "oracle" performance.

2. **PEBBLE**: The PEBBLE algorithm from Lee et al. [18], which does not use any prior data.

3. **Init**: This baseline demonstrates the importance of our adaptation procedure during training. Instead of re-adapting the reward model each time new feedback is collected, we initialize the reward model with the pretrained weights, and then perform standard updates with the Adam optimizer [60] as in PEBBLE.

For each environment, we reduce the total feedback budget by a factor of 20 in comparison to the maximum value used in PEBBLE. Full results are shown in Figure 2. Overall, we find that despite the $20\times$ reduction in feedback our method is able to solve almost all of the tasks with a near 100% success rate. In the Appendix, we directly compare to Lee et al. [18] with using their amount of feedback. In all tasks, except Button Press, we achieve the same asymptotic performance as SAC with $20\times$ less feedback than originally used for PEBBLE in Lee et al. [18] which is unable to learn a meaningful policy under a reduced feedback budget. In Appendix A we directly compare to PEBBLE with the feedback schedules from Lee et al. [18]. While the *Init* baseline generally performs better than PEBBLE, it still falls short of our method, indicating that re-adaptation is important. Unlike in other pretraining and finetuning paradigms, preference learning is done online, causing the optimal reward function induced by the data to shift. Re-adapting weights each time feedback is collected ensures that we get the full benefits of MAML by considering all data points. Locomotion experiments and ablations on feedback and query selection are included in the Appendix.

### 4.2 Can few-shot preference learning be used with humans?

While no human could be sensibly be expected to provide thousands of pieces of feedback, around a hundred or less not too daunting a task. Given the lower query-complexity of few-shot preference-based RL, we use it to learn complex robot manipulation policies from real-human feedback for the first time. In the process of doing so, we encountered a few challenges. First, humans often have a difficult time answering queries asked by preference based RL algorithms. Queries sampled by maximizing disagreement across an ensemble of reward functions often look identical to humans. Such queries at the margin may be maximally informative, but are more difficult to answer (See Figure 5). For example, it is unlikely that humans can accurately compare two behavior segments that only have slight variations in the robot joint positions. While this is not explicitly examined in prior work that largely uses artificially generated queries, it is important when considering the abilities of humans and our desired to adapt reward functions with a handful of data points, making everything more sensitive to errors.

To address this, we add the ability for human users to "skip" difficult queries instead of providing noisy answers and increased the number of uniform queries used to reduce the likelihood that difficult queries were presented. Second, despite these mitigations, humans still make mistakes in labeling resulting in query labels that are possibly inconsistent. We thus allowed policies to train for longer periods of time between feedback sessions in hopes of collecting more data on the current policy's belief over the reward.

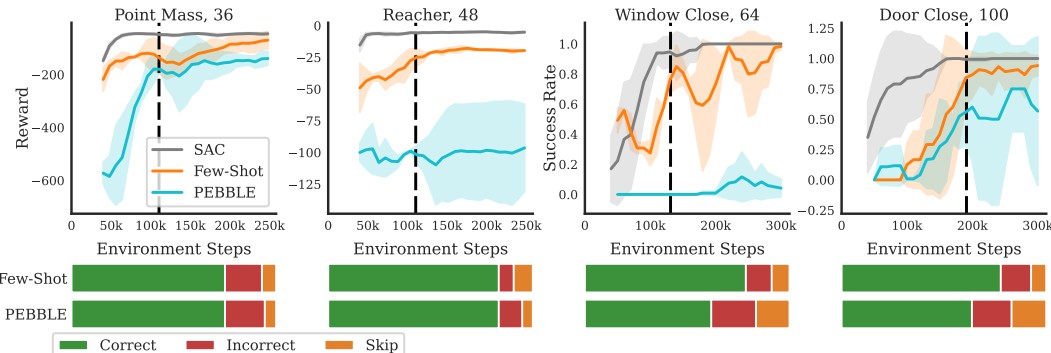

Figure 3: Results on training DM Control and Meta-World tasks from real human feedback. The vertical dashed black line indicates the point at which feedback was stopped. The horizontal bars at the bottom show the proportion of times users provided feedback that "correctly" agreed with the tasks ground-truth reward function, "incorrectly" disagree with the ground-truth reward function, or skipped the comparison.

After making the aforementioned changes, we examined the performance of few-shot preference-based RL on two of the MetaWorld environments and two additional environments based on the DM Control benchmark [61]. We take the point mass and reacher environments from DM Control [61] and change the reward function to be the negative L2 distance to an unknown goal. Reward models are pre-trained on random data and evaluated on unknown goal positions. The MetaWorld environments are as described in Section 4.1. Our full results are shown in Figure 3. As the ground-truth reward value for DM control correspond to the cumulative distance to the goal, the higher reward values of our method indicate that it can better communicate the human's objective with fewer queries. While PEBBLE was completely unable to solve Window-Close from human feedback, it made non-trivial progress on Door Close. This is likely because Door Close can be trivially solved by slamming the robot arm into the door instead of first grasping the handle and then closing the door as is encourage by our reward-function prior. Moreover, we find that in the Meta-World environments, users have an easier time answering queries from our method in comparison to PEBBLE. This is likely because the reward function prior guides agents towards interacting with objects, leading to more distinguishable behaviors. Results with more users on Reacher are in Appendix A.

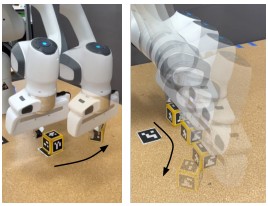

Figure 4: Push rollouts

| Goal Position | Reach | | Block Push | |
|---|---|---|---|---|
| | $(.55, .35)$ | $(.45, -.3)$ | $(.35, .3)$ | $(0.35, -0.3)$ |
| Few-Shot | $\mathbf{.061 \pm .041}$ | $\mathbf{.056 \pm .009}$ | $\mathbf{.188 \pm .175}$ | $\mathbf{.056 \pm .035}$ |
| PEBBLE | $.105 \pm .056$ | $.129 \pm .067$ | $.280 \pm .065$ | $.173 \pm .097$ |

Table 1: Results for the real-robot tasks. Performance is measured in meters to the desired goal position, lower is better. The $z$ targets for reach were 0.125 and 0.25, respectively. Results are averaged across multiple initial environment configurations. Best method is bolded.

### 4.3  Can Few-Shot preference-based RL be used in the real world?

Finally, we investigate the use of few-shot preference-based RL in real world settings using a Franka Panda Robot. We design two basic tasks: reaching and block pushing where the robot moves its arm or the block, respectively, to an unknown goal location communicated only via the learned reward function. We pre-train reward models with artificial queries and learn policies in simulation. We then transfer the learned policies to the real world and test on unseen goal locations. Table 1 contains our results. Performance is measured in meters to the true goal. Again, few-shot preference learning consistently outperforms PEBBLE despite the large sim-to-real gap. One additional benefit of our approach in real-world settings is that it potentially requires less instrumentation, as measurements previously needed to functionally compute reward are not required when using human feedback.

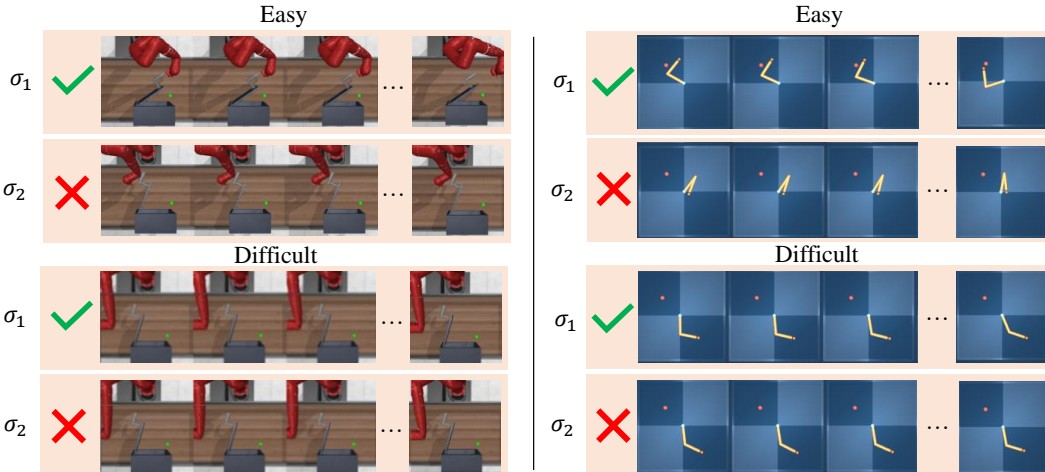

Figure 5: Examples of queries asked on the MetaWorld Door-Close and DM Control Reacher environments. In each figure the top segment, $\sigma_1$ had higher cumulative reward.

## 5 Discussion

**Limitations and Future Work.** While few-shot preference learning has several benefits, it has its limitations. Here we list the most salient ones, and possible means of overcoming them:

*Query Complexity*. Despite our gains in query-efficiency, the most complex tasks still require more feedback than we would like. Future work could examine how to expand the set of pre-training data.

*Pre-training Methods* . We investigate pre-training with artificial query data due to its generality, though our method could be used in combination with other pre-training objectives, like direct reward regression, to further boost performance.

*Pre-training Data*. While meta-learning methods have proven to be somewhat robust to changing dynamics in the real world [62], the efficacy of reward adaptation under larger distribution shifts induced by sub-optimal users, new tasks, or sim-to-real transfer remains in question. For example, if a new task is significantly out of distribution, we would expect training a reward function from scratch to perform better than adapting. Furthermore, pre-training can occasionally over-regularize the learned reward model, as exhibited in the Door Close experiment in Section 4.2.

*Query Difficulty*. Many queries asked by preference learning algorithms are too difficult for humans to answer, as shown in Figure 5. In fact, we find in Section 4.2 that active query schemes often result in queries that are too difficult for users to answer. Future work should explicitly consider how easy it is for a human to answer a query and not just its theoretical information content.

*User Inconsistency*. Unlike reward oracles, humans will inconsistently label queries. This challenge is only exacerbated when attempting to crowd source data from many users with differing styles. Future work can investigate additionally modeling human users.

**Conclusion.** We shift the paradigm of human-in-the-loop RL from the single-task to the multi-task setting, unlocking additional data sources that can be used to boost the query-efficiency of preference-based RL Algorithms. We believe our work's change in perspective to be a crucial stepping stone towards training robots with human feedback. Our novel few-shot preference-based RL method is able to effectively minimize the number of human queries required to train complex manipulation policies as demonstrated by our 20X improvement on standard benchmarks and effectiveness at real-human training.

**Acknowledgments**

This research was supported by NSF (1849952, 1941722, 2218760), ONR, and Ford. JH was supported by the Department of Defense (DoD) through the National Defense Science Engineering Graduate (NDSEG) Fellowship Program.

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
