# OpenReview forum: "Few-Shot Preference Learning for Human-in-the-Loop RL"
_robot-learning.org/CoRL/2022/Conference — CoRL 2022 Poster_

### Official Review · Reviewer_m37o · 2022-07-07

**Originality:** Good
**Technical Quality:** Good
**Clarity Of Presentation:** Good
**Impact:** 3

**Recommendation:**

Weak Accept: I recommend accepting the paper, but will not argue for my recommendation if the majority of other reviewers have a different opinion.

**Summary:**

This paper studies preference-based RL in a setting where the agent is also given to an offline dataset of preference/reward data from other tasks. The main idea is to use the offline dataset to meta-learn the preference model, which is then adapted on the target task. The motivation is that such meta-learning should significantly decrease the number of human queries required to solve the target task. Empirically, the proposed method outperforms 2 baselines (vanilla finetuning, no access to offline data) on 6/6 tasks. Additional experiments show that the method can work from real human feedback, and that the learned policy can be adapted to work on a physical robot.

**Issues:**

Larger issues:
* Many of the claims in the introduction require evidence or citations (see below)
* L160 - L164 -- I found this a bit confusing because it seems like providing a meta-learning dataset of preferences would be very, very expensive. Also, this seems a bit disingenuous, since none of the experiments actually involve meta-learning on real human preference data.
* L195 "paradigm shift" -- This seems to be overclaiming a bit.
* L249 -- L262 -- Why is a different environment used for these environments. This seems a bit suspicious, especially since the new DMC environments appear easier.
* L264 -- L268 -- Do humans ever actually provide preference feedback on real robot experience? The "sim2real" note seems to imply otherwise. This seems important to clarify.


Writing comments
* L12 "Contrary to ..." -- I found this explanation very helpful!
* "20X" --> "$20\times$"
* "Deep Reinforcement Learning" --> "deep reinforcement learning"
* "Markov Decision Process" --> "Markov decision process"
* "inordinate", "galvanized", "unduly" -- awkward word choice
* L28 -- I found the discussion about door opening with sparse rewards a bit disingenuous, as many prior works can learn such tasks with sparse rewards.
* L34 "aim to capture" -- Cite.
* L37 "their environment" -- Cite.
* L41 "learned reward functions are ..." -- Cite. (This claim seems surprising to me)
* L46 "seems hopeless" -- Cite, or provide evidence.
* L48 "... or linear reward models" -- Cite.
* L49 "necessitate expressive reward models." -- Cite.
* L53 "is far to" --> "is far too"
* L83 broken reference
* L86 "they are harder for users to provide" -- Cite.
* Fig 1 "it" -- Ambiguous pronoun reference
* L90 "These assumptions ... " -- Cite.
* L107 "classification" -- Is the implicit assumption here that reward learning is equivalent to classification?
* L119 -- What is $\mathcal{P}$?
* L123 "only that it ..." -- At this point, I was confused about what the prior data actually contained. Was it rewards, preferences, demos? (This is answered a few paragraphs later)
* L127 "multiple tasks" -- I was confused by this, as Fig 1 only seems to show adaptation to a single task.
* L129 -- The placement of $\pi_{new}$ makes it seem like $\pi$ is a task, not a policy.
* L133 "[50]" -- Does this 2016 paper really discuss meta-RL?
* L133 "model reward" --> "model the reward"
* L135 -- Why use trajectory segments instead of individual transitions?
* L177 "prior works" --> "prior work" (or, provide more citations)
* L210 "difficult ... as they are sequential" -- This seems a bit disingenuous, as most of these tasks can be solved in 10 - 20 steps.
* L261 "prior guides the agents" -- It'd be great to provide evidence for this claim.
* L445 -- L452 -- These citations have the wrong indent.

**Quality Of The Limitations Section:**

Limitations are addressed clearly

**Reviewer Expertise:**

4: The reviewer is confident but not absolutely certain that the evaluation is correct

**Robotics Focus:**

Sufficient demonstration on hardware

**Strengths And Weaknesses:**

Strengths
* The paper is generally well written.
* To the best of my knowledge, the proposed method is novel, and seems to address an important limitation with preference-based learning.
* The results show strong gains relative to the prior method (PEBBLE).

Weaknesses
* My main concern is with the central premise of the paper: that preference-based RL is an effective way to learn tasks. While there is a large and growing body of work that studies this problem setting, I am unaware of convincing prior work that shows that it is easier/cheaper for humans to provide preferences than other sorts of supervision. For example, in the DM Control tasks described in L251, specifying a goal (or, equivalently, reward function) requires just 2 numbers. I would expect that specifying the goal location wouldn't take more than a few seconds, of similar complexity to specifying a preference comparison. Similarly, doing back-of-the-envelope calculations for the results in Fig 2: The tasks can all be described by a hand + object goal state (3 + 3 dimensions, sometimes less). If we want 1/16 resolution, we need 4 bits / coordinate, for a total of 24 bits of supervision. In contrast, the proposed method uses at least 10k / 20 = 500 bits of supervision (21x).
* I found the presentation of the results in Fig 2 a bit confusing because the "20x" improvement isn't visualized anywhere. If possible, it'd be great to make a plot where one axis directly shows the 20x improvement (e.g., plot reward vs number of queries)
* The real-world results are a bit lackluster because, if I understand correctly, they aren't actually obtained by querying human users with real robot images but rather are obtained through sim2real. While this isn't an issue for most robotics papers, it does seems like a missed opportunity for a paper studying how robots should learn from humans in the real world.
* A number of minor writing comments (e.g., claims without citations attached). See below.

**Summary Of Recommendation:**

Overall, this paper is strong in many respects: it presents a novel method, is well written, and shows compelling empirical results. The paper does a great job delivering on the stated title ("few-shot preference learning for human-in-the-loop RL"). My main concern with the paper is that I'm not convinced that preference based learning is an effective method for soliciting human feedback. If the task can be described as reaching a simple goal state, it seems easier to ask the human to provide that goal state than to provide O(100s-1000s) preferences. While this paper doesn't purport to show that preferences are more efficient to provide than reward functions or goal states, I think that studying this question is critical for demonstrating the *relevance* of work; without this, it seems like this paper is a great contribution to an irrelevant problem. Concretely, there are a number of ways the paper might study the effectiveness of different sorts of supervision. One easy method would be to quantify how many seconds it takes a human to provide a goal state versus a preference, assuming that a reasonable UI is used for both. If the paper can show that preference-based learning is more effective than alternatives (e.g., manually-specifying the reward function or goal state), I would advocate for accepting the paper.

------------------------
**Update after rebuttal** -- The discussion period clarified and resolved most of my concerns about the paper. The additional experiments seem to provide good evidence for the claim that the proposed method can tackle tasks that are hard to define in terms of rewards or success states. I now vote for accepting the paper.

---

> ### Author Response · Authors · 2022-08-17
> **Response to Reviewer m37o**
>
> We would like to thank the reviewer for their extremely thorough job reviewing our paper. The comments left were very helpful in improving the draft! We were excited to hear that our paper was “strong in many respects” and that it was “novel”, “well written”, and “shows compelling empirical results”. The reviewers primary concern was the relevancy of the preference based-learning paradigm itself. We will first address the preference based learning concerns, then respond to the reviewers' individual questions. Due to space constraints, our response will be split into multiple comments.
>
> **Preference Based RL**
>
> The reviewer's primary concern is “with the central premise of the paper: that preference-based RL is an effective way to learn tasks.”  and they are not convinced that “prior work shows it is easier/cheaper for humans to provide preferences”. We respectfully disagree with the statement that preference based learning is not an effective way of learning tasks for the following reasons:
>
> 1. Reward design has proven to be extremely difficult [1,2,3,4]. Not only that, it is sometimes impossible to summarize everything into a single goal in many cases [1], such as safety vs comfort vs efficiency considerations when driving [5]. So we disagree with the statement that the tasks of interest can simply be performed by optimizing for a specified goal state. As an example, consider designing a reward function for the simple MetaWorld task Drawer Close: the reward can simply be 1 once the task is achieved, and 0 otherwise. On this exact task with a sparse reward function, SOTA model-based RL techniques like LatCo and PlaNet still only achieve a 46% and 22% success rate respectively after 500k environment steps [6]. Such tasks also do not have clear “achieved states”, rendering them inapplicable to relabeling techniques like HER [7] in the model-free setting. Moreover, policies trained on similar tasks in the sparse Androit benchmark, like door opening and hammering, attain worse than random reward 500k steps of SAC [8, Table 2] and are usually only solved with access to guiding optimal demonstrations [9]. Alternatively, we could have reward functions with better convergence properties that have individual terms for multiple agent, object, and goal distances, capture difficult to define concepts like grasping a handle, and notions of success. Such reward functions are non-trivial to design and depend on more than just a single goal state.
> 2. We would like to emphasize that designing reward functions is a major problem that has the potential to benefit from human feedback. However, we agree that a fair question to ask is if human preferences can provide the right signal for addressing this problem. We will discuss this next. If we agree that all desired behavior cannot be captured by simple hand-designed reward functions, we need a way of learning them. The reviewer is not convinced that preferences are an easy or effective way to do so. We sympathize with the reviewer’s skepticism, and agree with the fact that many prior preference based learning works were not impactful in robotics. However, we believe this is largely because they either a) assume simplistic reward models [10,11], which have shown to be inexpressive enough for complex robotics tasks [12], or b) they use neural networks but need tens of thousands of samples for training [13,14] which we absolutely agree is unrealistic and not cheap or easy. Our work addresses both of these drawbacks by ensuring we can rely on neural reward models while only needing a reasonable number of queries for learning.
>
> Our paper represents an opportunity to address the exact concern the reviewer has brought up. By shifting to the multi-task setting, we actually make it cheaper and easier to collect feedback from users. Because of that, we believe preference learning can be an effective way of learning rewards in regimes where writing reward functions is impractical. In many other settings when new tasks are being introduced, it may also be more efficient to learn from preferences than to hand design a new reward function.

---

> > ### Author Response · Authors · 2022-08-17
> > **Response Continued (Part 2)**
> >
> > **Bits of Supervision for MetaWorld**:
> > The reviewer characterizes the Meta-World tasks as only requiring “24-bits of supervision” using goal states. In practice, we find this to be far from the truth. In fact, reward shaping beyond simple “goal states” was critical in the development of Meta-World. The original **dense** rewards for MetaWorld which involved complex combinations of object, gripper, and goal distances were still insufficient to solve many of the tasks. Only once the authors spent significant effort designing new, carefully tuned reward functions for MetaWorld V2 could the tasks be solved (see https://groups.google.com/g/metaworld-announce/c/F9IixmpdL8M). For many of the tasks we use, goal states appear to be insufficient as algorithms like SAC were originally unable to converge to 100% success rates even with simple dense rewards. This also disregards any further arguments that could be made about learning efficiency under higher quality reward functions.
> >
> > **Time to Collect Human Preferences**:
> > The reviewer asked us to quantify the amount of time it took to collect human preferences. As we train policies interactively, we are unfortunately unable to disambiguate this for the existing human experiments. However, we know that it took our human user around 22 minutes in Point Mass, 28 minutes in Reacher, 45 minutes in Window Close, and 1 hour in Door close to provide all required feedback for four seeds in parallel. In the MetaWorld tasks in particular, where the ground-truth reward function communicates far more than just a goal location, 45m to an hour is a relatively low cost for obtaining four policies. If we measured just the human interaction time separately from training time, this value would likely be much lower. We have included more details on our human experiments in Appendix B.4. Though we cannot know for certain, this amount of time is likely lower than the amount of time that it took the MetaWorld authors to tune reward functions before attaining a similar level of performance.
> >
> > **Larger Questions**
> >
> > *Presentation of 20X results*: We have added a new version of Figure 2 in Appendix A.2 that plots the number of queries on the X-axis. We originally chose to plot environment steps for consistent comparison with baselines and to show the convergence behavior of our method after feedback stops. We also include an ablation on the amount of Feedback in Appendix A.1 that shows PEBBLE trained with the full amount of feedback, and our few-shot method with even less feedback. Again, we see that our method performs better than PEBBLE with 20 times less feedback, and in 3/6 environments performs better than PEBBLE with 40 times less feedback.
> >
> > *L160 - L164 Meta-Learning Datasets* This statement was designed to indicate the flexibility with which meta-learning datasets for our method can be generated and in practice, not all data needs to be generated from humans. In fact, some of our experiments use purely random interactions. Full details on the datasets can be found in Appendix B.
> >
> > *Sim2Real Results*: The reviewer is correct that the preferences for the real-robot experiments were elicited in simulation. We have clarified this in the text.The purpose of this experiment was to test if policies learned from preferences were robust enough to work in the real world, even when trained with limited data. Only policies trained with our method were effective at solving the tasks.
> >
> > *DM Control Environments*: The standard point mass and reacher environments in DM control have no notion of goals. In the original point mass environment, the agent is randomly initialized but reaches the same point (center) in every episode. In the original reacher environment, the location of the desired point is provided as part of the state space. We modified these environments as described in Appendix A in order to make them applicable to our setup, where we need a distribution of tasks, each with a different reward function.
> >
> > *Paradigm Shift* Reworded.

---

> > > ### Author Response · Authors · 2022-08-17
> > > **Response Continued (Part 3)**
> > >
> > > **Writing Comments**
> > > Thank you so much for the in-depth review of our work. We believe we have addressed all writing comments listed in the main text. Here we detail the significant changes and supporting references. Smaller changes can be found using blue text in the updated draft.
> > >
> > > * L28 Sparse Reward Door Opening: The majority of benchmarks (MetaWorld, RoboSuite) and real world environments [15] for door opening use dense reward signals. To our knowledge, door opening benchmarks with sparse reward do not make progress with regular SAC (see Androit Door Open task with Online SAC in Table 2 of [14]), and often use expert demonstrations [9].
> > > * L34 "aim to capture" - Combined with the next sentence to make it more clear we are referring to “Inverse Reward Design”  Hadfield-Menell et al.
> > > * L37 "their environment" - added reference on effects in scaling state and action spaces in reward hacking.
> > > * L41 “learned reward functions” - added citations on reward shaping with dense learned reward functions, models of aligning human intent with preferences, and adapting learned reward functions.
> > > * L48 Linear reward models: added several citations
> > > * L49 necesitate expressive reward models: added citation to [12] which we believe provides an exact example of what the reviewer is looking for – a robot golfing task where linear reward models are explicitly unable to express all goals.
> > > * L86 harder for users to provide: Cited psychology paper discussing how users are unable to rank for than two or three tones. Also, making a binary choice seems to be intuitively easier than rankings.
> > > * L90 These assumptions: again added reference to [12]
> > > * L133 "[50]" - This paper does not discuss Meta-RL, but it discusses the difficulties when optimizing RL objectives with deep nets.
> > > * L135 trajectory segments: added two citations showing the effectiveness of segments
> > > * L210 “Difficult.. Sequential”: The argument being made here is that tasks involving the manipulation of objects often involve multiple components, like grab the block and then move it. Because we are learning the reward online, we have to first learn the reward for grabbing the block before we can begin to learn the reward for moving the block. The reviewer also stated that MetaWorld tasks can be solved in 10-20 steps. Empirically, we did not find this to be the case . During training MetaWorld tasks are fixed to a horizon of length 500. We computed the average number of timesteps before the success condition was met over 10 episodes in all the environments we used with policies trained using SAC. They were 49.2 for Window Close, 54.2 for Door Close, 33.8 for Door Unlock, 41.4 for Button Press, 45.0 for Drawer Open, and 44.1 for Sweep Into.
> > > * L261 “Prior Guides the agent”: As stated in the text, this hypothesis is supported by the visualizations of the policies learned for the door opening task with human feedback, which can be found at the accompanying project website linked in the abstract.
> > >
> > > [1] “Quantifying Differences in Reward Functions” Gleave et al. ICLR 2021
> > >
> > > [2] “Avoiding side effects in complex environments” Turner, Ratzlaff, Tadepalli
> > >
> > > [3] “Preferences implicit in the state of the world”. Shah et al. ICLR 2019
> > >
> > > [4] “Concrete problems in ai safety” Amodei et al.
> > >
> > > [5] “The Green Choice: Learning and Influencing Human Decisions on Shared Roads” Biyik et al. IEEE CDC 2019
> > >
> > > [6] “Model Based Reinforcement Learning via Latent-Space Collocation” Rybkin & Zhu et al. ICML 2021
> > >
> > > [7] “Hindsight Experience Replay” Andrychowicz et al. 2017
> > >
> > > [8] “D4RL: Datasets for Deep Data-Driven Reinforcement Learning” Fu et al. 2020
> > >
> > > [9] “Learning Complex Dexterous Manipulation with Deep Reinforcement Learning and Demonstrations” Rajeswaran, Kumar et al. 2017
> > >
> > > [10] “Active preference-based learning of reward functions” Sadigh et al. RSS 2017
> > >
> > > [11] “Bayesian preference elicitation for multi-objective engineering design optimization” Leopard et al. Journal of Aerospace Information Systems 2015.
> > >
> > > [12] “Active preference-based gaussian process regression for reward learning” Biyik et al. RSS 2020
> > >
> > > [13] “PEBBLE: Feedback-Efficient Interactive Reinforcement Learning via Relabeling Experience and Unsupervised Pre-training” Lee*, Smith*, Abbeel ICML 2021.
> > >
> > > [14] “Deep Reinforcement Learning from Human Preferences” Christiano et al. NeurIPS 2017
> > >
> > > [15] “Dexterous Manipulation with Deep Reinforcement Learning: Efficient, General, and Low-Cost” Zhu, Gupta et al. ICRA 2019

---

> > > ### Comment · Reviewer_m37o · 2022-08-17
> > > **Reviewer response**
> > >
> > > Thanks for the incredibly detailed response! The new experiment in A.2 well-addresses my question about visualizing the 20x improvement, and the writing revisions address my concerns about unsubstantiated claims and clarity. The other questions/comments in the response address all my other concerns, except the one about relevance.
> > >
> > > > that preference-based RL is an effective way to learn tasks
> > >
> > > The response makes a great point here: that there are actually two notions of task specification. (1) how many questions/information is required to mathematically specify the task? and (2) how many queries are needed to learn the task using SAC (or another method)? My back-of-the-envelope numbers were for #1; I agree with the authors that #2 would be much larger, and is arguably a more relevant quantity to consider and optimize.
> > >
> > > One area where I disagree with the response is regarding the difficulty of specifying tasks via other means, and especially via goal states or successful states. For example, without using reward shaping or demonstrations, [1] learns a goal-conditioned policy for the metaworld drawer task that achieves a distance of 2cm after 500k steps (success is defined as $\le$4cm [2]). Hammering tasks have also been solved given only examples of successful states [3]. These papers provide some evidence that disputes the claims in the response: similar tasks do have clear success states, are amenable to relabeling techniques, do achieve rewards better than random, and can be solved without optimal demonstrations.
> > >
> > > [1] https://arxiv.org/abs/2011.08909, Figure 3.
> > >
> > > [2] https://github.com/rlworkgroup/metaworld/blob/master/metaworld/envs/mujoco/sawyer_xyz/v2/sawyer_drawer_close_v2.py#L10
> > >
> > > [3] https://arxiv.org/pdf/2103.12656.pdf, Figure 2

---

> > > > ### Author Response · Authors · 2022-08-18
> > > > **On Goal States, C-Learning, and RCE (Part 2)**
> > > >
> > > > **2. Scaling of Classifier Based Methods**
> > > >
> > > > Furthermore, we believe that these approaches will still potentially struggle as the task horizon and dimensionality grows. Both C-Learning and RCE rely on learning a classifier with the addition of a “bootstrapping” term. From our understanding, there is no free lunch, and both methods still need to collect data interacting with relevant parts of the state space near goal of success criterion for their variant of classifier-based bootstrapping or bellman update to occur. As state and action dimensions and horizon scales, the probability of collecting this experience from a randomly initialized policy becomes exponentially unlikely. While one might argue that the induced rewards in C-Learning and RCE are probabilities, and thus provide more learning signal than pure sparse rewards, such probabilities are only accurately estimated by the classifier in the support of the data collected, not the entire state and action space, and similar problems remain.
> > > >
> > > > 1. C-Learning circumvents some of this by using relabeling strategies which add additional environment constraints, like the ability to measure intermediate positions of the drawer, that may be impossible or ineffective in more complex scenarios. For example, if the drawer is placed at the edge of the robots operating space instead of near the grippers initialization, the probability of interacting with the drawer goes down substantially, and relabeling of the drawer position becomes less fruitful.
> > > >
> > > > 2. RCE does not use relabeling, but instead relies on many diverse, positive examples for the classifier to generalize. For example, RCE requires 200 examples in the MetaWorld tasks, which is on the same order of magnitude as our preference learning technique, albeit with pre-training data. Despite this, and the application of additional tricks in RCE like n-step returns, RCE still uses 3 million interactions instead of the 1 million we use.
> > > >
> > > > Classifier based learning, though more efficient than Q-learning in the chosen settings, still encounters many of the same challenges as sparse reward learning. On the other hand, reward functions learned from preferences are immediately dense and thus immediately guide policy learning. In this way, preference based reward functions do not have the same data collection problems that C-Learning and RCE might face under increasing dimensionality and horizon, as with preferences we can define dense rewards even without collecting any data near goal or success criterea.
> > > >
> > > > **3. Environment Comparison**
> > > >
> > > > The environments used in both C-Learning and RCE are modified to only have a horizon of length 151, instead of the standard 500 timstep horizon in MetaWorld that we use. See https://github.com/google-research/google-research/blob/master/rce/rce_envs.py#L61 and https://github.com/google-research/google-research/blob/master/c_learning/c_learning_envs.py#L131 . This non-standard choice in episode horizon makes exploration and learning substantially easier than in our experiments and we believe circumvents some of the issues discussed in the previous point. If the horizon was set to the usual 500 steps, we would expect C-Learning and RCE to be less performant.
> > > >
> > > > **4. Defining Rewards is still hard**
> > > >
> > > > Finally, we would like to re-emphasize that there are many tasks for which defining a success state or criteria may be difficult in the first place. This is evident in autonomous driving regimes where the tradeoff between safety and comfort is difficult for humans to express in a mathematical success criteria, but easier to infer from trajectory comparisons. For example if obstacles are detected on a road, a car might want to slow down for safety purposes, but doing so might be jerky and uncomfortable. In this scenario, a human designer would have a hard time defining a reward function or “goal state” that balances conflicting objectives, but would be able to specify their preferences through comparisons.
> > > >
> > > >
> > > > [1] “PEBBLE: Feedback-Efficient Interactive Reinforcement Learning via Relabeling Experience and Unsupervised Pre-training” Lee*, Smith*, Abeel ICML 2021.
> > > >
> > > > [2] “ROIAL: Region of Interest Active Learning for Characterizing Exoskeleton Gait Preference Landscapes”
> > > > Kejun Li, Maegan Tucker, Erdem Bıyık, Ellen Novoseller, Joel W. Burdick, Yanan Sui, Dorsa Sadigh, Yisong Yue, Aaron D. Ames
> > > > International Conference on Robotics and Automation (ICRA), May 2021
> > > >
> > > > [3] “Active Preference-Based Learning of Reward Functions” Dorsa Sadigh, Anca D. Dragan, S. Shankar Sastry, Sanjit A. Seshia Proceedings of Robotics: Science and Systems (RSS), July 2017

---

> > > > ### Author Response · Authors · 2022-08-18
> > > > **On Goal States, C-Learning, and RCE (Part 1)**
> > > >
> > > > Thank you so much for your quick response to our comments, we appreciate it! We are glad to hear that we have well-addressed your questions about visualizing the 20x improvement in query-efficiency, writing claims and clarity.
> > > >
> > > > We would also like to thank you for sharing a recent set of works that is able to converge on environments that are similar, but not the same (Point 3.). The provided publications propose different ways of circumventing reward functions in order to use goals (C-Learning) or examples (RCE). While we agree this direction is exciting and the works are indeed of high-quality, the approaches are currently not standard or general to all settings (Point 1.). The community largely uses accepted model-free and model-based RL techniques for which our preference based learning approach directly carry over. Thus, even under the context of these works we still maintain our position that preference based reward learning is a relevant problem to tackle, and provide more justification below.
> > > >
> > > > **1. Applicability of C-Learning and RCE**
> > > >
> > > > The provided references make the implicit assumption that the underlying reward being optimized is some form of state condition. For C-Learning it is the probability ratio $p(s_{t+1} = s_{t+} | s_t, a_t) / p(s_{t+})$ (Appendix E.1) and for RCE it is the probability of a state satisfying a given task (Section 4.1). While this problem setup is amenable to some tasks, it necessarily precludes objectives that cannot be “classified”. This is especially evident for tasks that have explicitly temporal objectives. In fact, many goal based methods theoretically make the assumption that the MDP terminates once the goal state is reached (C-Learning, Section 3), which explicitly precludes its use when we care about behavior over time. If human designers care about intermediate behavior and not just if a state satisfies a task, RCE is also rendered inapplicable. There are numerous examples of this in the real world:
> > > >
> > > > 1. Learning locomotion policies requires considering sustained reward over multiple timesteps. We would not consider an agent to be “running” if it achieves a particularly high velocity in just one state at one timestep. One could imagine redefining the MDP to include history, but doing so would drastically increase the dimensionality of learned classifiers.
> > > >
> > > > 2. Another example is autonomous driving, where we care about safety objectives over extended periods of time and not just the car’s final destination.
> > > >
> > > > While it is unclear how to adapt C-Learning or RCE to these settings, preference based RL techniques have been shown to work in both locomotion [1,2] and autonomous driving [3]. In this regard, C-Learning and RCE are not as broad as our approach, so comparing with them seems unduly limiting. In other words, to the best of our understanding, C-Learning and RCE implicitly make restrictive assumptions — we find this analogous to the prior work on “active preference based learning with linear reward”,  where the specific linear structure was limiting their general applicability. Instead, our work does not make any of these explicit or implicit assumptions and can be applied more generally than C-Learning and RCE.

---

> > > > > ### Comment · Reviewer_m37o · 2022-08-20
> > > > > **Reviewer response (to both Parts 1 and 2)**
> > > > >
> > > > > Thanks for the very detailed response! This is useful for clarifying my understanding of both the proposed method and the prior work.
> > > > >
> > > > > The argument about goal states being hard to specify in some settings (end of Part 2) makes a lot of sense to me; I find that convincing. Would it be possible to run the proposed method in that sort of setting (i.e., where defining the task in terms of goals or success states would be difficult)? I think that adding an experiment to that effect would go a long way in convincing readers that preference-based reward learning is practically important for realistic problems.

---

> > > > > > ### Author Response · Authors · 2022-08-22
> > > > > > **Additional Locomotion Experiments**
> > > > > >
> > > > > > We have added an experiment to Appendix A.3 in a setting where defining the task in terms of goal states is difficult. Please see the revised download in the top-level comments to view the updates. In particular, we use the Cheetah Velocity environment from https://arxiv.org/abs/1703.03400, with only 10 training tasks and a horizon of 500. In this environment, the agent is asked to run at different speeds. From our previous discussion, defining such temporal objectives like running etc. is difficult to define in terms of goal states.
> > > > > >
> > > > > > Our results are in line with the metaworld manipulation experiments: Our few shot method appears to converge after only around 100 queries, while PEBBLE is unable to attain close to the same performance, even with access to 1000 queries.
> > > > > >
> > > > > > We hope the experiment addresses your concern. Again, thank you so much for engaging with us!

---

> > > > > > > ### Comment · Reviewer_m37o · 2022-08-23
> > > > > > > **Reviewer response**
> > > > > > >
> > > > > > > Thanks for running this additional experiment! I appreciate the difficulty of getting new experiments running on such short notice.
> > > > > > >
> > > > > > > The environment in this example does provide some nice intuition for why defining tasks in terms of full goal states can be limiting. There are a few things that I'm unsure about here:
> > > > > > >
> > > > > > > * The task reward in this example is pretty simple, right? (For MAML, it's the L1 distance from the target velocity, plus an L2 control penalty [1])
> > > > > > > * Would goal-conditioned RL actually fail in this case? What if you provided a few examples of "running fast" states?
> > > > > > >
> > > > > > > [1] https://github.com/cbfinn/maml_rl/blob/master/rllab/envs/mujoco/half_cheetah_env_rand.py#L62

---

> > > > > > > > ### Author Response · Authors · 2022-08-26
> > > > > > > > **Comparing Directly with RCE**
> > > > > > > >
> > > > > > > > In order to better ground the discussion, we have run experiments with RCE in the Cheetah environment and a new sparse Point Mass-like environment with a middle obstacle. They can be found in Appendix A.4 of the updated paper.
> > > > > > > >
> > > > > > > > **Cheetah:**
> > > > > > > >
> > > > > > > > The Cheetah environment is designed to test the performance of example based methods to work in settings with simple temporal-based reward functions. The task reward is simple as the reviewer points out, but we use the simple nature of this task to tease apart the failure points of different methods. We find that relabeling states with the target velocity is insufficient for RCE to function, and it completely fails.
> > > > > > > >
> > > > > > > > If an expert demonstration is used to provide example states, RCE works but it is less sample efficient than our approach and converges to slightly lower performance. It makes sense that RCE can work with expert demos -- any states that look like the expert are considered as satisfying the task -- but this is besides the point: RCE was intended for settings where we do not have pre-existing expert data for the target task. It is unrealistic to assume that expert demonstrations can be provided before a policy is trained, particularly for locomotion policies on a different embodiment. In fact, we believe it is rather difficult for humans to determine "example states" in this environment, as they have little intuition on which particular joint position and velocities are good for running. Realistically, humans can select the desired velocity and specify states with it. We did this exactly by relabeling randomly collected states with the specified target velocity, and under these conditions we find that RCE completely fails. The temporal nature of the task makes it more difficult for humans to specify through examples. In less toy environments, we would expect this effect to be more pronounced.
> > > > > > > >
> > > > > > > > **Point Mass Barrier:**
> > > > > > > >
> > > > > > > > The Point Mass-like Barrier environment is designed to test the performance of example-based methods in very sparse reward settings. In this sparse setting, RCE is unable to obtain any reward signal and performs worse than our method and PEBBLE. We posit that this is because rewards learned from preferences are dense, while RCE still requires data to be collected near the examples for its bellman-like backup.
> > > > > > > >
> > > > > > > > We hope that these experiments provide both some intuition and evidence on why preference learning can be practically useful in some settings.
> > > > > > > >
> > > > > > > > We have included the code for our RCE experiments in the supplementary submission, which was directly using the RCE author's code.

---

> > > > > > > > > ### Comment · Reviewer_m37o · 2022-08-26
> > > > > > > > > **Reviewer response**
> > > > > > > > >
> > > > > > > > > Thanks for running these additional experiments. These seem to provide good evidence for the claim that the proposed method can tackle tasks that are hard to define in terms of rewards or success states. I will vote to accept the paper. [1]
> > > > > > > > >
> > > > > > > > > [1] I'll change the score in my review once the author discussion period ends.

---

### Official Review · Reviewer_FhLm · 2022-07-27

**Originality:** Fair
**Technical Quality:** Good
**Clarity Of Presentation:** Good
**Impact:** 2

**Recommendation:**

Weak Accept: I recommend accepting the paper, but will not argue for my recommendation if the majority of other reviewers have a different opinion.

**Summary:**

This paper introduces a few-shot preference learning framework via meta learning. A task in this setting is represented as solving for the optimal policy of an MDP with known transition but unknown reward function.

To learn the reward function, the model assumes access to a collection of preference indications over pair of state-reward trajectories. Each indication is 0/1 signaling whether a segment of state-reward trajectories is preferable to another. The reward function can then be learned via optimizing Eq. (1) which is the cross-entropy objective corresponding to the Bradley-Terry model of paired comparison.

To combat the scarcity of data in new task, the paper assumes access to preference data of prior (similar) tasks. A direct application of MAML on the above cross-entropy objective function thus enables learning a base parameterization for the reward function, which can be fine-tuned to new tasks with little preference data.

**Issues:**

I think the main issue with this paper is that it has not highlighted the non-trivialities that require new learning capabilities to be innovated to enable the application of MAML. It would be good if the authors can focus on this while preparing the rebuttal.

**Quality Of The Limitations Section:**

Limitations are addressed clearly

**Reviewer Expertise:**

3: The reviewer is fairly confident that the evaluation is correct

**Robotics Focus:**

Sufficient demonstration on hardware

**Strengths And Weaknesses:**

Strength:

The paper is generally well-written. The technical exposition is easy to follow.
Clear pseudocode of the proposed algorithm is also included which is nice.
Experiments with both simulation & hardware are presented, which has a strong robotic focus

Weakness:

The paper appears to be a direct application of MAML on an existing preference learning model but it is not quite clear what are the non-trivialities in doing so?

From a learning perspective, MAML is a model-agnostic recipe that can be applied to any loss functions that are optimizable via gradient descent/ascent so suppose that preference data is available for a range of relevant tasks, I tend to think that if there is nothing else, then the application here is straight-forward.

The extensive experiment of course requires significant effort but it does not seem to add much insights to the literature as the empirical questions it addressed are pretty much the same set of questions that were raised and reasonably addressed in MAML (as a general principle to pre-training) and its follow-up variants with various applications (robotic applications included).

I do notice an interesting point (4.2) about few-shot learning with human feedback. This might be a new aspect that has not been explored much but I do not see how this is connected to the technical formulation. In fact, the narrative in 4.2 is mostly about the potential noisiness of human feedback and engineering feature to allow human to skip on hard queries.

It might be more interesting to develop learning policies that seek human feedback in an optimal way, e.g. how to structure queries that are less ambiguous to human & given limited interaction budget, what are the most informative queries; and whether such proactive query strategies would reduce the sample complexity & etc.

--

These are my current understanding of this paper (from a learning perspective) but please do correct me if I miss anything important. I will be happy to discuss.

**Summary Of Recommendation:**

This paper seems to be a direct application of MAML on an existing preference learning model. From what was presented in the main text, I unfortunately do not see much new insights (as MAML is a generic model-agnostic recipe for pre-training).

The application with real robotic scenario is nice & interesting but on its own, it is only a re-confirmation to the broad applicability of MAML with no new insight. With this, I believe this paper does not quite meet the novelty bar for publication.

---

> ### Author Response · Authors · 2022-08-17
> **Response to Reviewer FhLm**
>
> We would like to thank the reviewer for their feedback. We were encouraged to hear that our paper was “well-written”, and look forward to engaging with the reviewer. Due to space constraints, our response will be split into two parts. The reviewer’s primary concern was with the novelty of our proposed method in relation to MAML. While from a pure learning perspective we did not change MAML itself, from a broader perspective we believe that our work has significant value to the community.
>
> First, our method does indeed leverage MAML for active preference-based learning, but our problem formulation is novel and important for advancing human-in-the-loop techniques. Prior methods for active preference learning were extremely limited because they either relied on overly simplistic reward models that are unable to capture the complexities of real-world tasks, or had complex neural models that required tens-of-thousands of user queries no human would be willing to answer. Our new perspective on the preference learning problem is to leverage past data to alleviate these issues. To our knowledge, we are the first to consider the multi-task preference based RL setting. Though our method does combine MAML with preference learning techniques, doing so advances the state of preference based robotics by actually allowing humans to teach robots in a feedback-efficient manner, which simply was not possible in prior work. This is evidenced by our 20X improvement in feedback efficiency in comparison to PEBBLE.
>
> Second, the reviewer views the simplicity of our method to be a potential drawback, to which we respectfully disagree. Methods with greater simplicity are often the most important for creating real robotic systems, as they are generally easier to implement and more likely to have widespread adoption. Thus, we view the simplicity of our method as a benefit and not a drawback. That being said, we do agree that simpler approaches may require a higher bar of evaluation to merit publication. In that regard, the majority of reviewers agree that our experiments are strong: “results back up the intended claims of the paper” (Reviewer mqZu), and “show strong gains relative to the prior method” (Reviewer m37o). During the rebuttal period, we have also shown that in several environments our method can be even more query efficient through new ablations found in Appendix A.1. We believe our results demonstrate that our approach can have a significant impact on the field by allowing for learning from human preferences in realistic robotics settings with neural rewards.
>
> Finally,  several other works that integrate MAML into different problem settings and show strong improvements have been accepted to top-tier venues, including CoRL. We believe just because they have integrated two known ideas, that doesn’t mean these works are not valuable as long as they present a new way of addressing an important existing problem. The impact of these works is also evident by their citation counts. For example, [1] (CoRL) directly applies MAML to learning reward functions from goal images, [2] applies MAML to adapt dynamics functions, and [3] (CoRL) directly applies the MAML objective to imitation learning datasets. In that regard, our work leverages the benefits of MAML in a similar manner to alleviate issues in preference based learning which for many years has suffered from poor query-complexity.

---

> > ### Author Response · Authors · 2022-08-17
> > **Response Continued**
> >
> >
> > Aside from the novelty question, the reviewer brought up two additional points.
> >
> > **Considerations when Learning from Human Feedback:**
> > The reviewer thinks that it would be interesting to further develop algorithms that give humans easier questions to answer, and we wholeheartedly agree! We discuss considerations for learning from real human feedback in Section 4.2, and would like to point out that our work is already a step towards addressing these challenges. We find that altering feedback schedules, including “skip” queries, and guiding questions via pre-training to be important design decisions. Moreover, we only encountered the additional complexities of learning from real humans because we were actually able to learn robot policies from humans due to the improved query efficiency of our method. In this way, our work opens the door to many exciting follow ups outside the direct scope of our paper as discussed in Section 5.
> >
> > **The Non-trivialities when Applying MAML to Human Feedback:**
> > The reviewer asked for more information on extra considerations when applying MAML in the active preference learning setting. This is a great point, and our approach addresses these in the following ways:
> > The init baseline demonstrates that it is important to reset the weights of the pretrained network when collecting new data in an active setting (Section 4.1)
> > Adding “skip” queries and adjusting schedules when learning from humans. (Section 4.2)
> > Increasing the number of initial uniform queries for human users to ask fewer questions right at the margin. (Section 4.2)
> > Changing the feedback schedule to allow policies to train longer under the noisiness of human labels (Section 4.2).
> >
> >
> > [1] “Few-Shot Goal Inference for Visuomotor Learning and Planning” Xie, Singh, Levine, Finn CoRL 2018.
> > [2] “Learning to Adapt in Dynamic, Real-World Environments through Meta-Reinforcement Learning” Nagabandi et al. ICLR 2019
> > [3] “One-Shot Visual Imitation Learning via Meta-Learning” Finn & Yu et al. CoRL 2017

---

### Official Review · Reviewer_pAzu · 2022-07-29

**Originality:** Good
**Technical Quality:** Good
**Clarity Of Presentation:** Fair
**Impact:** 2

**Recommendation:**

Weak Accept: I recommend accepting the paper, but will not argue for my recommendation if the majority of other reviewers have a different opinion.

**Summary:**

The overall idea of the paper is:
1. Use an existing dataset to meta learn a reward function that can quickly adapt to a few comparison tuples $(\sigma_1, \sigma_2, y)$. The reward function is used in the preference prediction, and the underlying meta learning method is MAML.
2. Given a new task, as well as a few comparison tuples, adapt the reward function, then perform RL. Specifically, the comparison tuples are provided every $K$ iterations, where the most "uncertain" segments are selected as queries for human.

**Issues:**

Please clarify the points I mentioned in the weakness section or provide further empirical evidence.
In general, I think adding more ablations on important design choices of the algorithm can much improve the paper. Minor point: Fig.2 will look better if its resolution is higher.

**Quality Of The Limitations Section:**

Limitations are addressed clearly

**Reviewer Expertise:**

3: The reviewer is fairly confident that the evaluation is correct

**Robotics Focus:**

Relevant but unlikely to deploy to hardware in near future

**Strengths And Weaknesses:**

***Strengths***:
1. To my knowledge, the overall idea is novel and straightforward: meta-learn the preference is not only reasonable, but potentially important for applying preference learning to real-world problem.
2. The "active learning" part also makes sense to me.
3. The overall presentation is clear and easy to follow.

***Weaknesses***:
1. It might be good to see an ablation over $K$ or essentially how the model performs with different shots.
2. It will also be good to see the difference of picking the $(\sigma_1, \sigma_2)$ that maximizes $std(P(\sigma_1 > \sigma_2))$ versus for instance randomly selecting two $\sigma$s. Basically how important it is to have this particular selecting scheme.


**Summary Of Recommendation:**

I think overall adding the meta-learning part to preference learning is a novel and important step. This is the main reason I recommend acceptance of this paper.

---

> ### Author Response · Authors · 2022-08-17
> **Response to Reviewer pAzu**
>
> Thank you for your time in reviewing our draft. We were delighted to hear that the reviewer agrees that the meta-learning component of our method is “important for applying preference learning to real-world problem[s]”. The reviewer primarily requested that additional ablations be run. We agree that additional ablations can increase our understanding of the method, and consequently have run them. Below we detail how we have added the requested experiments.
>
> **Ablation over K:** We are not entirely certain what the reviewer means by an ablation over K. In Section 3 and Algorithm 1, K refers to the frequency at which feedback is collected, and not “different shots” as mentioned in the review. Please let us know if ablations on the amount of feedback is what you meant. We have assumed as such and provided additional ablations as described in the next paragraph. If this is not what you meant, please let us know and we would be happy to provide other experiments!
>
> To better understand the effects of the amount of feedback used, we have made multiple improvements to the draft. First, we ablate the amount of feedback used in our few-shot method by training policies in MetaWorld with half the number of queries. For direct comparison with PEBBLE, we also train policies with the original amount of feedback used in Lee et. al. [1]. Results for this ablation can be found in Figure 6 in Appendix A.1. We find that in several environments, we can further reduce the amount of feedback given to our few-shot method by a factor of ½ and still attain similar performance. Additionally, we have created versions of Figures 2 and 3 which plot the amount of feedback on the X-axis, instead of the environment steps. These plots, which show performance scaling with feedback, can be found in Appendix A.2.
>
> **Ablation on Disagreement Sampling:** We originally did not include ablations on disagreement sampling as they have been done in prior work (Figure 5 of [1] and Figure 6 of [2]) and we faced space constraints. Nonetheless, we agree that the effect of active learning procedures may change in limited data regimes, and we thank the reviewer for bringing this to our attention. Ablations on the query sampling scheme can be found in Appendix A.1. We find overall that disagreement sampling helps performance in many environments, particularly Drawer Opening, but is not critical in others.
>
> As a side note we also completely redid Figure 2, and it is now a vector graphic.
>
> If we have addressed all the reviewers' listed weaknesses, we would kindly ask that they consider revising their score.

---

> > ### Comment · Reviewer_pAzu · 2022-08-26
> > **Response to Authors**
> >
> > Thank you very much for providing the additional ablation study on the amount of feedback. Yes, originally I was referring to how $K$ and $M$ affect the learning performance, as they are the important hyper-parameters of the algorithm. Now I think the additional experiment in the updated Appendix has resolved my concern. Moreover, thanks for providing the result in Fig. 7, which shows the benefit of the proposed uncertainty-based sampling method (though it will be good to further analyze why certain environment shows much more benefit).
> >
> > Based on the authors' response, I think the problem this paper considers is interesting and potentially important (as preference is weak supervision, which is much easier to provide). The solution proposed is reasonable and straightforward, supported by solid experimental results. Although there are papers showing how sample-efficient RL can be used to solve the MetaWorld problems, I think what this paper presents is orthogonal. In other words, if preference is the only supervision signal, how can the agent learn in a sample-efficient way from this signal. So this work can serve as a simple yet good baseline for future study of few-shot preference learning in robot learning tasks.
> >
> > To this end, I recommend acceptance of this paper.

---

> ### Author Response · Authors · 2022-08-26
> **Thank you again for the review!**
>
> Hi, we believe we ran the requested experiments and included them in the updated paper. As the discussion period is coming to a close soon, we may be short on time to run an "ablation over $K$", if the amount of feedback was not what was intended. Please do let us know.
>
> Thank you again!

---

### Official Review · Reviewer_mqZu · 2022-08-06

**Originality:** Very Good
**Technical Quality:** Good
**Clarity Of Presentation:** Very Good
**Impact:** 4

**Recommendation:**

Weak Accept: I recommend accepting the paper, but will not argue for my recommendation if the majority of other reviewers have a different opinion.

**Summary:**

This paper addresses the problem of learning end-to-end reward functions from preferences. Rather than rely on hand-defined feature representations, the proposed method learns a reward function directly from high-dimensional input. To avoid requiring excessive training data, the proposed method leverages meta-learning to learn over multiple tasks and quickly adapt to a new one using a tractable number of preference queries.


**Issues:**

I would love to see alternative graphs for Figures 2 and 3 that demonstrate the relationship between # of queries and performance, if you have them available. If this is not possible, any other discussion about this relationship would be useful.

Please do list the requested details about the technical approach and user study design.

Additional clarification:
"For each environment, we reduce the total feedback budget by a factor of 20 in comparison to the maximum value used in PEBBLE, but keep all other hyper-parameters the same."
Does this mean that the numbers in the Figure 2 graph titles are the number of queries used by the proposed method, but not PEBBLE?


**Quality Of The Limitations Section:**

Limitations are addressed clearly

**Reviewer Expertise:**

5: The reviewer is absolutely certain that the evaluation is correct and very familiar with the relevant literature

**Robotics Focus:**

Sufficient demonstration on hardware

**Strengths And Weaknesses:**

Strengths:
+ The approach is novel and described clearly.
+ The results back up the intended claims of the paper: the proposed method results in significantly (20x) fewer queries than current baselines.
+ The proposed method is impactful in making meta-learning approaches more tractable for interactive robot systems by making use of a well-studied interaction type (i.e., preferences).

Weaknesses:
- There is very little information provided about the user study that aims to validate whether this system is practical. For example, how many queries were posed to participants? How many participants were there? What was their background and experience with robots? How much time did the interactions take? What subjective measures were collected about how "practical" the interaction was? How were preferences presented to users? I understand that there may not be room for all of this information in the main paper, but it should be at least present in the appendix.

- Figures 2 and 3 present compelling results about the system's ability to converge during training. However, these graphs would be more interesting with a different x-axis. Rather than report performance w.r.t. # of update steps (with a fixed number of preference queries), it would be more interesting to see performance w.r.t. # of preference queries (with a fixed number of update steps). This would provide more granular information about how many queries are really needed by the system, and how much "faster" it learns compared to baseline (where faster is measured w.r.t. user engagement, rather than convergence speed).

- The network architecture also appears to be missing from the main paper and the appendix.


**Summary Of Recommendation:**

Overall, this is an interesting paper that presents techniques that will be useful to other researchers in interactive robot learning. The paper would be made even more impactful by adjusting the graphs to use a different x-axis (highlighting the relationship between # of queries and performance) and including necessary details on the approach and user study.

---

> ### Author Response · Authors · 2022-08-17
> **Response to Reviewer mqZu**
>
> We would like to thank the reviewer for their in-depth comments and review of our draft. We were excited to hear that they found our approach to be “novel and described clearly” and that our experiments “backed up the intended claims of the paper”. Below, we address the reviewers comments one-by-one.
>
> **Information on the User Study:**
> Our human experiments were designed to test if few-shot preference learning could be effectively used by a human expert to train policies. We would like to emphasize that we intentionally did not refer to this as a user study, because it was intended to be a usability test and did not explicitly test the response of multiple users to our system. Regardless, we have added a new section to the Appendix, section B.4, that provides all the requested details which we summarize here.
>
> For consistency we used one human subject which was familiar with preference based RL and the metaworld and DM Control benchmarks. Experiments were completed on Point Mass, Reacher, Window Close, and Door Close in that order. In order to reduce the burden on the human subject, all four trials for an environment were run in parallel. As feedback was elicited intermittently through the course of training, we cannot fully separate the time it took for users to answer queries with the time used to train the policy. However, we know that the total time before all queries were answered was around 22 minutes for Point Mass, around 28 minutes for Reacher, around 45 minutes for Window Close, and around 1 hour for Door Close. We measured the practicality of the user interactions by the number of skip queries as shown in Figure 3. The human user rarely needed to skip queries, particularly for the Few-Shot method indicating that the system is indeed practical to use. Users were shown images like the ones depicted in Figure 5 and those in Appendix D. From the figures in Appendix D, we see qualitatively that the questions asked by our few-shot method were easier to answer for a human user than those asked by PEBBLE.
>
> **Figures 2 and 3 with Different X-axis:**
> We agree that showing Figures 2 and 3 with feedback on the X-axis can reveal more information about the performance of different methods. We have included these versions of the plots in Appendix A.2. We originally chose to display environment steps on the x-axis, as it is the usual choice in preference based RL [1, 2, 3] and allows for the most similar comparison to prior work. Additionally, showing environment steps demonstrates the convergence properties of each method because, as shown in Figure 3, training continues after feedback stops. Nonetheless, plotting feedback shows the same trend: our few-shot method achieves higher performance in all environments. If the reviewer feels strongly about replacing the plots in the main body of the paper with the ones using Feedback on the X-axis, we can do so.
>
> Moreover, we ran additional ablations on the amount of Feedback in Appendix A.1. In this ablation, one can clearly compare our method to PEBBLE with the original amount of feedback (20 times ours), and can see that our few-shot method is still performant on several environments with even less feedback.
>
> **Network Architecture:**
> We used the same network architectures as in PEBBLE [1], and originally included the network sizes for all experiments in Appendix C in Tables 2 and 3. We have now emphasized this in Section 4. The reward networks use 3 Dense layers of size 256 with ReLU activations and a final Tanh activation. The actor and critic in SAC use ReLU activations and 2 or 3 Dense layers of size 256 depending on the environment.
>
> **Clarification on Feedback Amounts:**
> The number listed on the graph title is the amount of feedback used in total by all methods, except for SAC which acts as a ground-truth. What we mean by that statement is that if the original PEBBLE paper used 50,000 queries for a given task, we give all methods a budget of at most 50,000 / 20 = 2,500 queries. For a fair comparison, this feedback budget is the same across all methods in Figures 2 and 3. The full list of the schedules can be found in Table  4 in Appendix C. We also now include a direct comparison to PEBBLE with its original amount of feedback in Appendix A.1 as previously mentioned to further support our performance improvement claim.
>
> We have updated the draft accordingly (completely new ablations in Appendix A.1, new plots on feedback in Appendix A.2, user study in Appendix B.4), and in light of this ask the reviewer to consider raising their score.
>
> [1] “PEBBLE: Feedback-Efficient Interactive Reinforcement Learning via Relabeling Experience and Unsupervised Pre-training” Lee*, Smith*, Abbeel ICML 2021.
>
> [2] “Deep Reinforcement Learning from Human Preferences” Christiano et al. NeurIPS 2017
>
> [3] “BPref: Benchmarking Preference Based Reinforcement Learning” Lee at al. NeurIPS 2021 Dataset Track

---

> ### Author Response · Authors · 2022-08-26
> **Thank you again for the review.**
>
> Hi! As the discussion period is ending shortly, and we have added a number of experiments, we wanted to make sure we had time to address any remaining concerns.
>
> Thank you!

---

> ### Author Response · Authors · 2022-08-28
> **More user study information**
>
> As this is the end of the discussion period, we would just like to highlight that we have run more experiments across different human users in Appendix A.5. We hope the reviewer finds this interesting per the first point in the original review.

---

### Author Response · Authors · 2022-08-17
**Updated Paper with Revisions, New Ablations, and Figures**

**Comment:**

Thank you to everyone for reviewing our paper. We have responded to each reviewer individually, but make overarching comments here. Please refer to the updated paper and Appendix attached below.

**Feedback Ablations**: Several reviewers wanted to more clearly see the 20X improvement in query efficiency. We have included a new section ablating the amount of feedback for our method and PEBBLE in Appendix A.1, which clearly shows our method performing better than PEBBLE with 20 times fewer queries. We also show that our method remains performant on 3/6 environments after reducing the amount of feedback even further.

**Figures 2 and 3**: Several reviewers mentioned changing the axis on Figures 2 and 3 to show feedback. We have done this in Appendix A.2.

**Human Experiments** Many reviewers asked about the human experiments or the ease of collecting feedback. We have added a new section of the Appendix, B.4., giving more specifics on the human experiments, including the time it took before all feedback was given.

Other improvements to the draft were additionally made, and all changes can be shown in blue. The appendix in particular has been largely revised. Thank you so much to everyone!

**Zip File:**

/attachment/82d3d2144b2d00ab2d5a62066f706692afbd8c42.zip

---

### Author Response · Authors · 2022-08-22
**Updated Paper with Locomotion Experiments**

**Comment:**

We have added additional experiments in a Locomotion environment to Appendix A.3. The experiments show similar results as in manipulation -- our method converges with only about 100 queries, while PEBBLE does not attain close to the same performance with 1000 queries.

**Zip File:**

/attachment/446e613c5cc4e5933dcfcaa56a79d24747bf6d51.zip

---

### Author Response · Authors · 2022-08-26
**Updated Paper with Example-Based Experiments**

**Comment:**

In response to reviewer m370 questions we have added experiments directly comparing to the example driven method, RCE in Appendix A.4. The accompanying discussion summarizes many of the points made during the discussion period.

In response to reviewer Fhlm we have also made some changes to the main text that emphasize extra considerations when pretraining and adapting in the online preference learning setting.

For convenience, we have directly attached the appendix to the pdf this time.

Thank you everyone for a lively discussion!

**Zip File:**

/attachment/bee5703da2ab98fb819f836d6ab6ac2b990b6e32.zip

---

### Author Response · Authors · 2022-08-27
**More human results**

**Comment:**

Over the past week we have also studied the effect of varying human users for the Reacher experiment with human feedback in Appendix A.5. We hope that this helps bring to focus more of the challenges of adapting reward functions with human preferences, as Reviewer FhLm suggested. We believe this would also be of interest to reviewer mqZu.

See the updated draft attached below. Our experiments directly show the relationship between the percentage of the time that human users provide preferences in agreement with the ground truth reward function, and the resulting ground-truth reward of the final policy.

**Zip File:**

/attachment/3d0bc339d8b9e624978b1ab3fae1a6baca83f3ac.zip

---

### Meta-Review · Area_Chair_HZfw · 2022-08-06

**Recommendation:** Accept (Poster)
**Confidence:** 5

**Metareview:**

There is consistent agreement from all 4 reviewers that this is a well-motivated and well-written paper with contributions that could provide clear impact in robot learning. The proposed meta-learning approach is novel and straightforward with experimental validation both in simulation and on real hardware.

**Best Paper Nomination:**

No